# Optical inactivation of intracellular molecules by fast-maturating photosensitizing fluorescence protein, HyperNova
Hisashi Shidara[1], Taku Shirai[1], Ryohei Ozaki-Noma[2], Susumu Jitsuki[1], Takeharu Nagai [2] &
Kiwamu Takemoto [1] ✉

Photosensitizing fluorescence protein is a promising tool for chromophore-assisted light inactivation (CALI) that enables specific oxidation and inactivation of intracellular molecules. However, a commonly used monomeric photosensitizing fluorescent protein, SuperNova, shows a low CALI efficiency due to its insufficient maturation at 37 °C, thereby limiting the application of CALI to various molecules, especially in mammalian cells. Here, we present a photosensitizing fluorescence protein, HyperNova, with markedly improved maturation at 37 °C, leading to greatly enhanced CALI efficiency. Exploiting this quality, HyperNova enables the application of CALI to variety of molecules such as a mitotic kinase and transcriptional factors that were highly challenging with conventional SuperNova. To further demonstrate the utility of HyperNova, we have also succeeded in developing novel CALI techniques for MAP kinases by HyperNova. Our findings suggest that HyperNova has the potential to expand the molecular toolbox for manipulating biological events in living cells, providing new avenues for investigating cellular signaling pathways.

Optical manipulation techniques have attracted much interest for elucidating the causal relationship between protein dynamics and their physiological functions in spatiotemporal manner. Among them, chromophore-assisted light inactivation (CALI) is a promising optical method for local and acute inactivation of target proteins with light irradiation[1]. CALI uses photosensitizers, which generate short-lived reactive oxygen species (ROS) when light is absorbed, to oxidize and inactivate target molecules[2]. Because the diffusion radius of the ROS produced by this process is very short, approximately 1–4 nm[3,4], CALI can be expected to achieve specific molecular inactivation. To date, synthetic small molecules and genetically encoded proteins have been reported as photosensitizers for CALI experiments[5]. In particular, fully genetically encoded photosensitizing fluorescent proteins have attracted much attention in recent years because they can be easily fused with a target molecule to perform CALI experiments.

The first fully genetically encoded photosensitizing fluorescent protein was KillerRed[6]. However, since KillerRed forms a dimer, it often presents problems with abnormal localization when fused with target molecules. SuperNova, a monomeric form of KillerRed, improves on this point and

allows the manipulation of several molecules[7]. Furthermore, in a recent study, the application of SuperNova to mice in vivo revealed a part of the memory consolidation mechanism during sleep[8]; therefore its applications are expected to expand to in vivo studies in the future. However, we have noticed that when SuperNova is fused to a target molecule, the fluorescence of SuperNova is often weak, and it does not mature sufficiently to perform the CALI experiments in many cases. In addition, a previous report has suggested that SuperNova could be a molecule with low folding efficiency[9]. This problem could potentially be the main reason the CALI method has been applied only to a limited number of molecules. Therefore, solving the problem of low maturation efficiency in SuperNova at 37 °C should enable the manipulation of many molecules that could not be controlled by CALI thus far.

Here, we present HyperNova, a photosensitizing fluorescent protein with a markedly improved maturation efficiency at 37 °C. HyperNova has a much higher maturation efficiency at 37 °C than SuperNova, allowing high CALI activity early after expression begins. We also found that HyperNova matures sufficiently at 37 °C even when fused with target molecules in living cells. This property of HyperNova makes it possible to manipulate various

[1]Department of Biochemistry, Mie University Graduate School of Medicine, Mie, 514-8507, Japan. [2]SANKEN, Osaka University, Ibaraki, Osaka, 567-0047, Japan.
✉e-mail: takemotk@med.mie-u.ac.jp

types of molecules, such as a mitotic kinase and transcriptional factors, which were hard to inactivate by SuperNova. To further prove the utility of HyperNova, we have successfully developed a novel CALI method for MAP kinases, JNK1 and ERK2 by using HyperNova. Based on these results, we expect that HyperNova will allow us to manipulate many more molecules in the future, which will open the door to a variety of cell biological applications.

## Results

### Mutagenesis screening of HyperNova

First, by employing error-prone PCR and DNA shuffling, we introduced random gene mutations in SuperNova. After transformation of *E. coli* with the expression vector encoding the SuperNova mutants, colonies were formed on the PVDF membrane. Next, the membrane was transferred to an IPTG-containing plate to induce expression of the mutant proteins at 37 °C, and the fluorescence was observed at an early stage of expression when SuperNova fluorescence was not yet observable. After obtaining the plasmid from the colony that showed the strongest fluorescence, this process was repeated again. After a total of five screenings, the mutant SuperNova1.5 was obtained with a significant improvement in maturation compared to SuperNova at 37 °C, especially at 2.5 h after +IPTG, an early stage of expression (Supplementary Fig. 1a, 2.5 h: $p < 0.001$, 24 h: $p < 0.01$). The mutant contained 8 mutations (S10T/S79I/D114G/H148R/F150L/H152Y/M173V/D237Y) for amino acid substitution and 3 silent mutations relative to the SuperNova sequence.

When fused with fluorescent proteins, ferritin is known to reduce the maturation of fluorescent proteins. This property has been used in genetic screening to improve the maturation efficiency of fluorescent proteins when fused with other molecules[10]. In the second step of the screening, we fused the ferritin gene to SuperNova1.5 with a random mutation, expressed it in *E. coli*, and performed the same expression screening as above at 37 °C. After

four screening processes, a mutant SuperNova2.3 was identified with remarkable improvement in maturation efficiency compared to Super-Nova1.5 at both 2.5 h and 24 h after +IPTG (Supplementary Fig. 1b). SuperNova2.3 has 13 mutations (S10T/S79I/D114G/Q129L/H148R/M149I/F150L/H152Y/M173K/K202E/T207P/V225G/I228N) that cause amino acid substitutions and 3 silent mutations relative to the SuperNova sequence.

To quantify the maturation efficiency of SuperNova2.3 at 37 °C, we performed a comparative expression experiment at 37 °C in *E. coli* (Fig. 1a). The S10R mutant of SuperNova (SN-S10R) has been reported to show enhanced maturation at 37 °C[9]. Thus, we induced the expression of these three proteins in *E. coli* at 37 °C and examined the time course analysis for the intensity of their fluorescence in bacterial colonies to compare their maturation. We confirmed that SuperNova2.3 showed markedly improved maturation rate compared to SuperNova and SN-S10R (Fig. 1a). Despite the introduction of 13 mutations, the oligomeric state of SuperNova 2.3 was identical to that of the monomeric protein, SuperNova (Supplementary Fig. 2a). We therefore named SuperNova2.3 HyperNova and used it in subsequent analysis.

### Improved ROS production and CALI efficiency in HyperNova

HyperNova has an absorption spectral peak at 579 nm (Supplementary Fig. 2b). Its molar extinction coefficient is 37,500 $M^{-1}$ $cm^{-1}$. Spectroscopic analysis revealed that HyperNova has an excitation spectral peak at 570 nm and a fluorescence spectral peak at 593 nm (Fig. 1b), with a relative quantum yield of 0.30, which is equivalent fluorescence properties to SuperNova[7]. These results suggest that the ability of ROS production in HyperNova is not changed from SuperNova, and that only the maturation efficiency of HyperNova may have been improved. To verify this possibility, we tested the ROS generation upon light irradiation in purified proteins in vitro and in living mammalian cells.

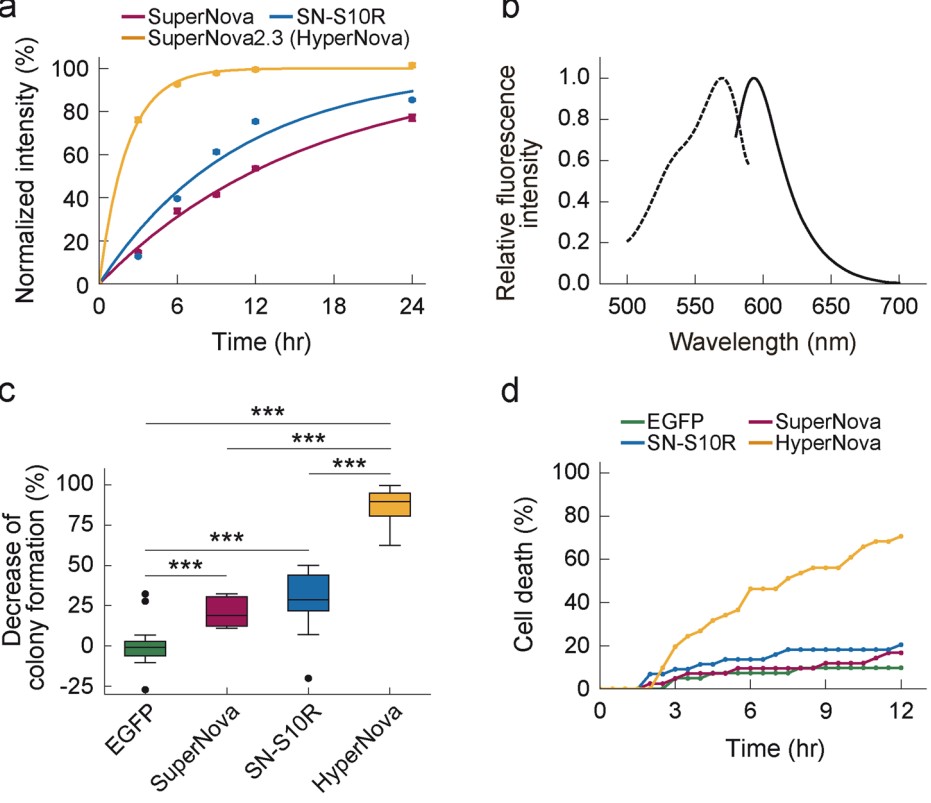

**Fig. 1 | Comparative analysis of maturation and CALI efficiency in SuperNova variants. a** Fast maturation of HyperNova protein in *E. coli* at 37 °C. The protein expression of SuperNova variants was induced by IPTG at 37 °C. The fluorescence of the colony was converted to the percentage of the maximum fluorescence intensity calculated by the fitting of a single exponential curve (n = 79 colonies for SuperNova and SuperNova2.3, $n = 125$ colonies for SuperNova-S10R (SN-S10R) from three independent experiments). **b** Excitation (dotted line) and fluorescence (solid line) spectra of HyperNova. **c** HyperNova has enhanced CALI efficiency in *E. coli* at 37 °C. CALI experiments in *E. coli* expressing SuperNova variants at 37 °C. The decrease in the number of colonies as a function of light irradiation is shown (vs. non-irradiation (%), $n = 16$ trials for mSEGFP, $n = 16$ trials for SuperNova, $n = 16$ trials for SN-S10R and $n = 22$ trials for HyperNova). $p < 0.001$ in EGFP vs. SuperNova, EGFP vs. SN-S10R, EGFP vs. HyperNova, SuperNova vs. Hyper-Nova and SN-S10R vs. HyperNova and $p = 0.345$ in SuperNova vs. SN-S10R by Wilcoxon rank sum test with Bonferroni correction. **d** HyperNova exhibits remarkably high CALI efficiency in mammalian cells. Time course analysis of cell death induction by light irradiation in mammalian cells overexpressing SuperNova variants in mitochondria. Data are plotted as the percentage of living cells versus time after light exposure ($n = 41$ cells for mSEGFP, $n = 42$ cells for SuperNova, $n = 44$ cells for SN-S10R and n = 41 cells for HyperNova). See also the representative images in Supplementary Fig. 4a. **p < 0.01 and ***p < 0.001, significant difference.

First, when the production of superoxide was measured in purified proteins that had matured sufficiently in vitro at 23 °C for 4.5 days, HyperNova was slightly less capable of producing superoxide (61.0 ± 1.79% in SuperNova, 57.3 ± 0.67% in SN-S10R, 53.0 ± 0.73% in HyperNova, $p < 0.01$ in SuperNova vs. HyperNova and $p = 0.128$ in SN-S10R vs. HyperNova, Supplementary Fig. 3a). Furthermore, when superoxide production was analyzed in living cells expressing photosensitizing fluorescent proteins at 37 °C, HyperNova was comparable to SuperNova (Supplementary Fig. 3b). On the other hand, when total ROS production is measured in living cells expressing photosensitizing fluorescent proteins at 37 °C, the production of total ROS in HyperNova with light irradiation was approximately 2 times higher than that in SuperNova and SN-S10R (Supplementary Fig. 3c, 0.34 ± 0.06 in SuperNova, 0.37 ± 0.05 in SN-S10R. 0.66 ± 0.08 in HyperNova). This suggests that the marked enhancement of the total ROS production in HyperNova at 37 °C is due to ROS other than superoxide. However, since the results from living cells include both the effects of folding ability at 37 °C and ROS production ability, it is difficult to determine which is improved in HyperNova.

To clarify this issue, we performed comparative experiments on the bleaching of the photosensitizing proteins in living cells at 37 °C. The bleaching of fluorescent molecules is caused by the production of ROS. In the bleaching experiment, only fluorescent molecules, that is, folded molecules, can be analyzed, so it is possible to determine whether total ROS production ability itself has certainly increased regardless of the difference of the folding ability between HyperNova and SuperNova. As a result, it was found that bleaching was slightly less likely to occur with HyperNova than with SuperNova (Supplementary Fig. 3d). Based on these results, we concluded that the marked increase in the total ROS production by HyperNova in living cells at 37 °C is due to the enhanced maturation of HyperNova and predicted that highly efficient CALI experiments will be possible even in mammalian cells.

Next, we analyzed the expected improvement in the CALI efficiency of HyperNova at 37 °C. A common method for comparing CALI efficiency in photosensitizing fluorescent proteins is to quantify their ability to induce cell death by light irradiation in *E. coli* and mammalian cells[6,7,11]. First, three photosensitizing fluorescent proteins were expressed in *E. coli* and cultured at 37 °C. At the early stage, 24 h post-expression, the bacterial suspension was irradiated with light to examine CALI efficiency through its ability to induce cell death. The results showed that HyperNova was able to perform CALI at more than three times higher efficiency than SuperNova and S10R (Fig. 1c). Moreover, a similar result was also observed in mammalian cells overexpressing HyperNova in the mitochondria. The time lapse imaging after light irradiation revealed that HyperNova had high CALI efficiency even when SuperNova and SN-S10R had no CALI effect (Fig. 1d and Supplementary Fig. 4a). No effect of CALI was observed in EGFP-expressing cells in these two experiments (Fig. 1c, d). This implies that the irradiated light itself had no effect on the experimental system, because EGFP does not absorb at the wavelength used. In addition, the CALI effect of HyperNova was also observed to be dependent on the light intensity (Supplementary Fig. 4a, b). These results indicate that HyperNova has high CALI efficiency from the early stage of expression due to its increased maturation efficiency at 37 °C.

## The improved maturation and CALI efficiency of HyperNova in fusion proteins

A particularly important experiment in the CALI method is the optical inactivation of a target protein that is fused with a photosensitizing fluorescent protein. We have frequently encountered the problems with SuperNova where the fusion molecule is hardly visualized, especially when expressed at 37 °C, probably due to low maturation efficiency of SuperNova. In the process of developing HyperNova, a second step of screening was performed to improve its maturation efficiency even in the ferritin-fusion molecule. Therefore, we expected HyperNova to mature sufficiently regardless of the fusion partner. To test whether the maturation efficiency of HyperNova was improved in the fusion molecule, the maturation efficiency

of photosensitizing fluorescent proteins was evaluated by analyzing the fluorescence intensity ratio to the amount of fusion protein quantified by immunohistochemistry (Fig. 2 and Supplementary Fig. 5). First, when HyperNova fused with c-jun and p53, it showed approximately 3-fold and 4-fold higher fluorescence intensity than SuperNova, respectively, in terms of the ratio to protein amount (274.0 ± 11.60% in c-jun-HyperNova vs. c-jun-SuperNova (Fig. 2a, d), 390.6 ± 17.46% in p53-HyperNova vs. p53-SuperNova (Fig. 2b, d)). In addition, fusion with Aurora A showed an approximately 2-fold increase in the fluorescence intensity ratio (189.0 ± 8.56% in Aurora A-HyperNova vs. Aurora A-SuperNova, Fig. 2c, d), and similar effects were observed with other molecules, including Actinin, Aurora B, Cyclin A, Cyclin D1 and Cyclin D2 (170.8 ± 10.62% in Actinin-HyperNova, 280.4 ± 17.65% in Aurora B-HyperNova, 176.0 ± 6.75% in Cyclin A-HyperNova, 304.8 ± 14.49% in Cyclin D1-HyperNova, 385.2 ± 17.43% in Cyclin D2-HyperNova, compared with SuperNova fusion proteins, Supplementary Fig. 5). Based on the fluorescence quantum yield (0.30 in SuperNova and HyperNova) and molar extinction coefficient (28,400 $M^{-1}$ $cm^{-1}$ in SuperNova, 35,400 $M^{-1}$ $cm^{-1}$ in HyperNova at 571 nm) of the purified protein, HyperNova was estimated to have about 1.25 times higher fluorescence intensity than SuperNova. For all of the molecules analyzed in this study, the fluorescence intensity ratios in HyperNova vs. SuperNova are much higher than this. Therefore, the present results considered to be due not only to the high fluorescence intensity of HyperNova, but also to the high maturation efficiency of HyperNova than that of SuperNova.

To evaluate whether HyperNova actually increased the CALI efficiency of the fusion molecule, we employed c-jun, p53 and Aurora A as target molecules of CALI. First, we tested the CALI of c-jun and p53 which were fused with HyperNova or SuperNova. To compare CALI effects in both transcriptional factors, we examined luciferase reporter assay with or without light irradiation using reporter plasmid encoding a luciferase gene downstream of the binding sequences of each transcriptional factor (Fig. 2e, f). In HeLa cells expressing c-jun-HyperNova fusion protein, c-jun transcriptional activity was clearly suppressed by light irradiation. In contrast, the SuperNova fusion protein did not induce the inactivation of c-jun with light (Fig. 2e). In addition, similar results also were observed in CALI experiment of the p53 fusion protein (Fig. 2f). Next, we also tested the CALI of Aurora A, a serine-threonine kinase involved in mitosis, whose inhibitors have been shown to induce mitotic arrest and cell death[12]. To compare CALI efficiency between SuperNova and HyperNova, each fusion molecule with Aurora A was expressed in HeLa cells, and their CALI efficiency was analyzed by the extent to which CALI induces mitotic arrest or cell death in HeLa cells by live imaging. The results showed that light irradiation of each fusion molecule induced 41.2% cell death in HyperNova and 16.7% in SuperNova (Fisher's exact test, $p = 0.4805$ in SuperNova and $p < 0.05$ in HyperNova). Furthermore, the time to cell division was also greatly prolonged in HyperNova (7.82 ± 1.573 h in HyperNova, 2.01 ± 0.613 h in SuperNova and 1.15 ± 0.360 h in GFP, Fig. 2g, h), indicating a significant inactivation of Aurora A by HyperNova. In contrast, the SuperNova fusion protein did not induce any CALI effect in Aurora A. From the above, consistent with the results of c-jun and p53, inactivation of Aurora A was demonstrated only by HyperNova. (Fig. 2g, h). In conclusion, these results indicate that HyperNova has higher maturation efficiency than SuperNova even when fused with other molecules, leading to marked improvements in CALI efficiency.

Moreover, when HyperNova was fused to variety of molecules, including tubulin and actin, the fusion proteins were found to exhibit physiologically correct localization (Supplementary Fig. 6). Based on these results, it was suggested that HyperNova could be applied to various molecules and these results motivated us to develop new CALI methods for a further variety of molecules.

## Development of CALI technology for JNK1 using HyperNova

To further demonstrate the utility of HyperNova for variety of molecules, we attempted to develop novel CALI methods for MAP kinases, important

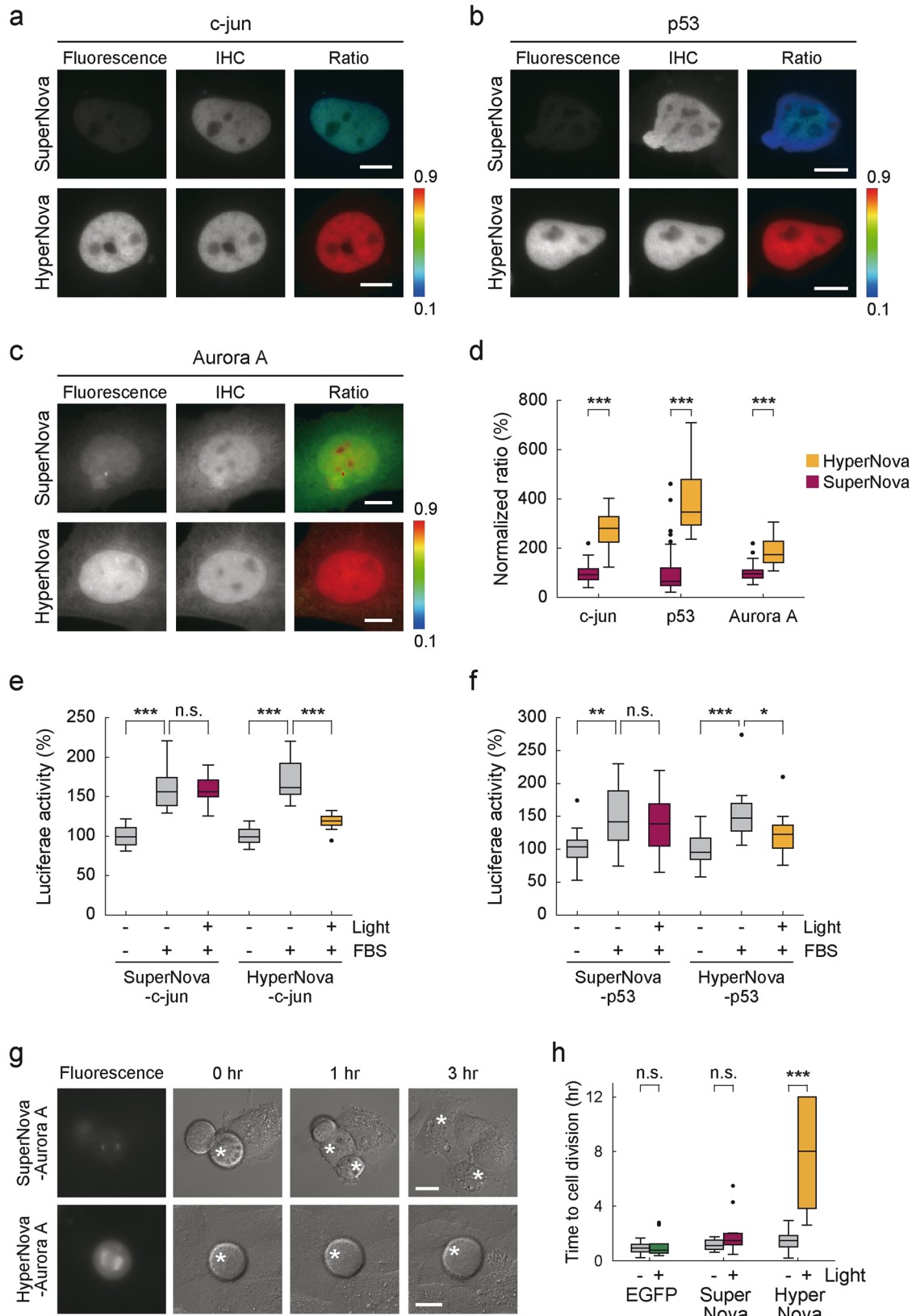

signaling molecules. JNK1 is a MAP kinase involved in intracellular signaling related to early development and the stress response[13], and ERK2 is involved in cell proliferation and migration[14]. In previous studies, visualization of JNK and ERK activity has been possible with FRET indicators[15], but optical inactivation of both molecules has not been reported thus far.

Therefore, if these molecules can be manipulated using HyperNova, it will strongly support the utility of HyperNova as a potential technology.

First, we expressed HyperNova fused with JNK1 in HeLa cells and analyzed whether light irradiation could inactivate JNK1. In this experiment, JNK1 activity was induced by the addition of anisomycin, and its

**Fig. 2 | Comparative analysis of maturation and CALI efficiency in fusion proteins. a–c** Marked improvement of HyperNova maturation in fusion proteins. HeLa cells expressing c-jun (**a**), p53 (**b**) and Aurora A (**c**) fusion proteins were subjected to immunostaining with an anti-KillerRed antibody and Alexa488-conjugated secondary antibody. The ratio image was calculated by the fluorescence of SuperNova or HyperNova divided by that of Alexa488 to detect fluorescence intensity per recombinant protein molecule. IHC indicates immunohistochemistry. The ratio images are shown in pseudocolor. Bar indicates 10 μm. **d** The average ratio of individual cells is shown (n = 50 cells for HyperNova-p53, n = 48 cells for Super-Nova-p53, n = 45 cells for c-jun-HyperNova, n = 43 cells for c-jun-SuperNova, n = 49 cells for HyperNova-Aurora A and n = 48 cells for SuperNova-Aurora A). $p < 0.001$ for p53, c-jun and Aurora A by the Wilcoxon rank sum test. **e** HyperNova enables CALI of c-jun. Transcriptional activates of c-jun were detected by luciferase promoter assay with or without light irradiation to induce CALI in HyperNova or SuperNova fusion proteins (n = 12). The activity of c-jun was normalized by the average in control experiment (light-/FBS-). The activity of c-jun was not changed by light irradiation in SuperNova but was decreased in HyperNova under light irradiation. $p < 0.001$ in light-/FBS- vs. light-/FBS + , $p = 1.000$ in light-/FBS+ vs. light + /FBS+ for SuperNova, and $p < 0.001$ in light-/FBS- vs. light-/FBS + , in

light-/FBS+ vs. light + /FBS+ for HyperNova by unpaired t test with Bonferroni correction. **f** HyperNova allows CALI of p53. Transcriptional activates of p53 were detected by luciferase promoter assay with or without light irradiation in HyperNova or SuperNova fusion proteins (n = 15-6). The activity of luciferase was normalized by the average in control experiment (light-/FBS-). $p < 0.01$ in light-/FBS- vs. light-/FBS + , $p = 1.000$ in light-/FBS+ vs. light + /FBS+ for SuperNova, and $p < 0.001$ in light-/FBS- vs. light-/FBS + , $p < 0.05$ in light-/FBS+ vs. light + /FBS+ for Hyper-Nova by Wilcoxon rank sum test with Bonferroni correction. **g, h** HyperNova enables CALI of Aurora A. **g** Comparison of CALI efficiency in SuperNova and HyperNova in Aurora A fusion proteins. The representative images for suppression of cell division were shown. **h** Cell division arrest by CALI of Aurora A with HyperNova. As described in the text, similar to the effects of previously reported inhibitors, cell death and prolonged division time were observed with CALI. In this graph, data were analyzed only for cells that were alive 12 h after light irradiation, and the time to complete cell divisions in individual cells is plotted (n = 10 cells for each). If cell division did not start within 12 h, the time was recorded for 12 h. $p = 1.000$ in mSEGFP, $p = 0.240$ in SuperNova and $p < 0.001$ in HyperNova by Wilcoxon rank sum test. Bar indicates 20 μm. *$p < 0.05$, **$p < 0.01$, and ***$p < 0.001$, significant difference. n.s. indicates not significant.

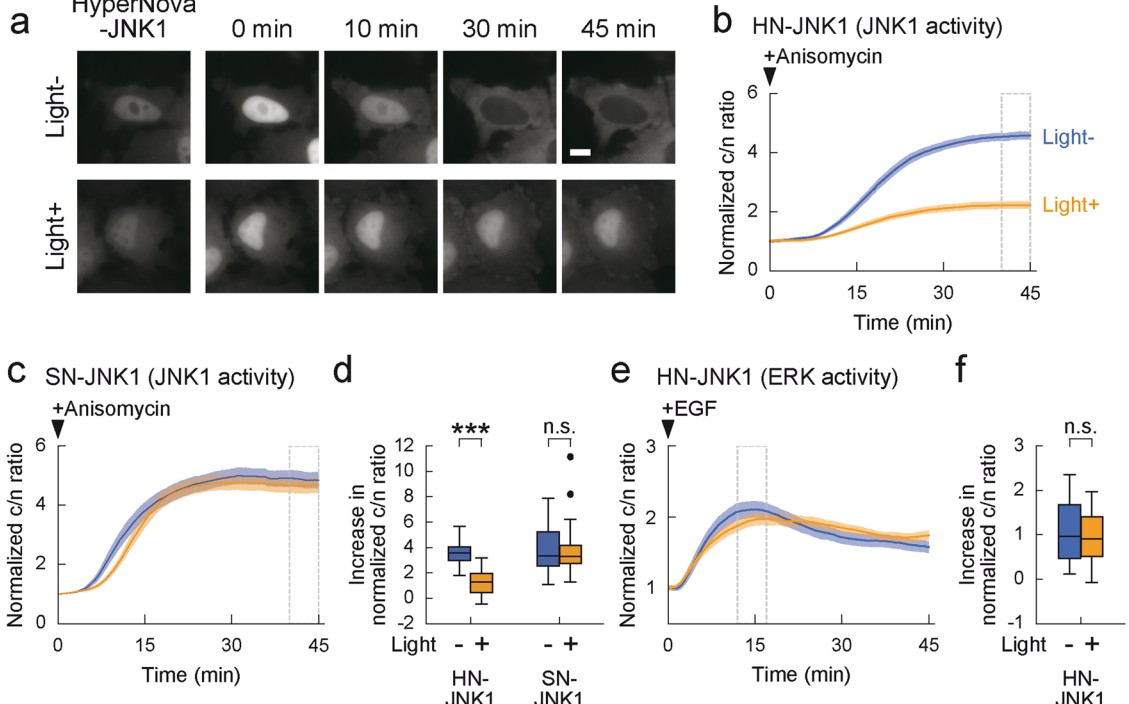

**Fig. 3 | Optical inactivation of JNK1 by HyperNova. a, b** HyperNova inactivates JNK1 by CALI. **a** The time course experiment of HeLa cells expressing JNK-KTR-GFP and HyperNova-JNK1 (HN-JNK1) with or without light irradiation after anisomycin stimulation. After addition of anisomycin, GFP images were taken at 15 s intervals for 45 min. Bar indicates 10 μm. **b** The average ratio of cytoplasmic to nuclear GFP fluorescence intensity with or without light irradiation was plotted versus time after anisomycin stimulation in individual cells expressing JNK-KTR-GFP and HN-JNK1. The ratio was normalized to the average ratio at 0 min (n = 47 cells for light- and n = 41 cells for light + ). **c** Experiments for SuperNova in individual cells expressing JNK-KTR-GFP and SuperNova-JNK1 (SN-JNK1) (n = 43 cells for light- and n = 49 cells for light + ). **d** HyperNova enables CALI for JNK1,

which is hard for SuperNova. The average increase in JNK-KTR-GFP signals from 40 to 45 min (dotted boxes in **b, c**) compared to the signals at 0 min. $p < 0.001$ in HN-JNK1 with JNK-KTR-GFP by unpaired t test and $p = 0.640$ in SN-JNK1 with JNK-KTR-GFP by Wilcoxon rank sum test. **e, f** Evaluation of the molecular specificity of JNK1 CALI by HyperNova. **e** Experiment in individual cells expressing ERK-KTR-GFP and HN-JNK1 to test ERK2 activity as negative control (n = 40 cells for light- and n = 46 cells for light + ). **f** Average increase in ERK-KTR-GFP signals at 12 to 17 min (dotted boxes in **e**) compared to the signals at 0 min. $p = 0.377$ by Wilcoxon rank sum test. ***$p < 0.001$, significant difference. n.s. indicates not significant. Data points in (**b, c, e**) are shown as the mean±s.e.m.

activation was visualized by the previously reported kinase translocation reporter (KTR)[16]. In this technique, the substrate and GFP are incorporated into the indicator; when the substrate is phosphorylated by kinase, the indicator migrates from the nucleus to the cytoplasm, allowing quantification of the enzymatic activity of kinase by comparing the ratio of GFP fluorescence intensity between the cytoplasm and nucleus. We irradiated cells expressing the HyperNova-JNK1 fusion molecule with light and

observed the effect of CALI by KTR. We found that the enzymatic activity of JNK1 was decreased in a light-dependent manner (HyperNova-JNK1, Fig. 3a, b, d, 3.56 ± 0.12 in light- and 1.22 ± 0.15 in light + ). In contrast, the CALI effect was not observed with SuperNova (SuperNova-JNK1, Fig. 3c, d, 3.87 ± 0.27 in light- and 3.66 ± 0.24 in light + ) or without HyperNova (JNK1 only, Supplementary Fig. 7a, b, 3.32 ± 0.20 in light- and 3.53 ± 0.12 in light + ), suggesting that HyperNova enabled for the manipulation of

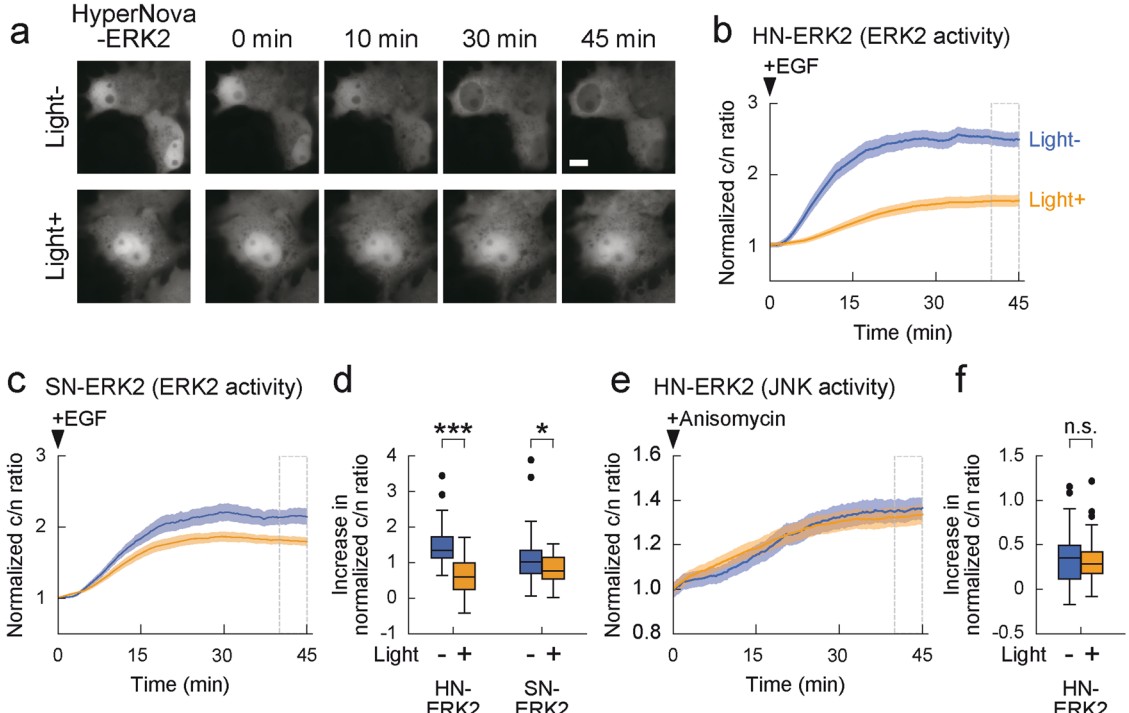

**Fig. 4 | Optical inactivation of ERK2 by HyperNova. a, b** HyperNova inactivates ERK2 by CALI. (**a**) The time course experiment of COS7 cells coexpressing ERK-KTR-GFP and HyperNova-ERK2 (HN-ERK2) with or without light irradiation after EGF stimulation. After addition of EGF, GFP images were taken at 15 second intervals for 45 min. Bar indicates 10 µm. **b** The average ratio of cytoplasmic to nuclear GFP fluorescence intensity with or without light irradiation was plotted versus time after EGF stimulation in individual cells expressing ERK-KTR-GFP and HN-ERK2. The ratio was normalized to the average ratio at 0 min ($n = 41$ cells for light- and $n = 42$ cells for light + ). Data points are shown as the mean±s.e.m. **c** Experiments for SuperNova in individual cells expressing ERK-KTR-GFP and SuperNova-ERK2 (SN-ERK2). ($n = 46$ cells for light- and $n = 49$ cells for light + ).

**d** HyperNova offers twice as high CALI efficiency as SuperNova in ERK2. Average increase in ERK-KTR-GFP signals at 40 to 45 min (dotted boxes in **b, c**) compared to the signals at 0 min. $p < 0.001$ in HN-ERK2 with ERK-KTR-GFP and $p < 0.05$ in SN-ERK2 with ERK-KTR-GFP by Wilcoxon rank sum test. **e, f** Evaluation of the molecular specificity of ERK2 CALI by HyperNova. **e** Experiment in individual cells expressing JNK1-KTR-GFP and HN-ERK2 to test JNK1 activity as negative control ($n = 50$ cells for light- and $n = 50$ cells for light + ). **f** Average increase in JNK-KTR-GFP signals at 40 to 45 min (dotted boxes in **e**) compared to the signals at 0 min. $p = 0.572$ by Wilcoxon rank sum test. ***$p < 0.001$, significant difference. n.s. indicates not significant. Data points in (**b, c, e**) are shown as the mean±s.e.m.

molecules that was completely impossible with SuperNova. To further verify the molecular specificity of CALI, we also checked ERK activity as a negative control for JNK1 CALI and found that ERK activity was not changed (Fig. 3e, f). These results indicate that the CALI method using HyperNova enables specific inactivation of the JNK1.

### Development of CALI technology for ERK2 using HyperNova

We next proceeded to develop a novel CALI method for another MAPK, ERK2. As mentioned above, no method has been developed to induce optical inactivation of ERK2 to date. As in our approach to manipulating JNK1, we irradiated COS7 cells expressing the HyperNova-ERK2 fusion molecule with light to induce CALI. After adding EGF, we observed the effect of CALI by quantifying ERK activity using ERK-KTR and compared it to the activity without light irradiation. The results showed that HyperNova effectively inactivated ERK activity (58% reduction, Fig. 4a, b, d, $1.50 \pm 0.10$ in light- and $0.63 \pm 0.08$ in light + ), and its efficiency was approximately 2-fold higher than that of SuperNova (30% reduction, Figs. 4c, d, $1.15 \pm 0.10$ in light- and $0.81 \pm 0.06$ in light + ). In contrast, the CALI effect was not observed in the absence of HyperNova fusion (ERK2 only, Supplementary Fig. 7c, d, $1.07 \pm 0.08$ in light- and $1.08 \pm 0.07$ in light + ). To further examine the molecular specificity of CALI for ERK2, we monitored JNK activity using KTR for JNK as a negative control and found no effect on JNK activity (Fig. 4e, f). These results indicate that ERK2 can be specifically and effectively inactivated by CALI with HyperNova.

### Discussion

More than 30 years after the CALI method was developed, the number of molecules that could be manipulated was still limited[5]. CALI using an antibody against the target molecule has the advantage of inactivating endogenous molecules and is a highly specific method used in vivo[17,18]. On the other hand, this method requires the preparation of antibodies capable of CALI, limiting its throughput and ease of implementation. In contrast, a genetically encoded monomeric photosensitizing fluorescent protein, SuperNova, has recently emerged. This has made it possible to express target molecules fused with SuperNova in cells and easily perform CALI. As a result, the number of molecules that can be manipulated by CALI is gradually increasing[5,8]. However, as shown in Fig. 2 and Supplementary Fig. 5, even monomeric SuperNova cannot highlight the target molecule in many cases, especially when the fusion molecule is expressed at 37 °C. This led us to hypothesize that conventional SuperNova could have low maturation efficiency at 37 °C, especially when fused with a target molecule. Based on this hypothesis, we performed expression screening for SuperNova mutants at 37 °C. The HyperNova obtained in this study actually showed distinctive superiority over SuperNova in living cells, suggesting that our hypothesis seems to be correct. The three new mutations (H148R/M149I/F150L) in HyperNova were located in the β-barrel, specifically at the interface in the KillerRed dimer[19,20]. When KillerRed was monomerized, the exposure of this surface to solvents could have destabilized SuperNova. The other mutations within the β-barrel (M173V/K202E) were also located in the vicinity of these three mutations. Thus, these mutations may have overcome the problems caused by monomerization of KillerRed, resulting fast maturation.

As reported in this study, total ROS production in living mammalian cells was increased by approximately 200% in HyperNova (Supplementary Fig. 3c). We also confirmed that photobleaching of HyperNova tends to be occurred rather lower than that of SuperNova in living cells. This suggests that the improved total ROS production at 37 °C in HyperNova results not from an improved ability of ROS production, but from an improved maturation at 37 °C. Based on these results, we conclude that HyperNova has a higher folding efficiency, resulting in a higher percentage of molecules folding in the cell, which leads to an increase in apparent ROS production especially at 37 °C. Therefore, it is convincing that the fluorescence quantum yields were comparable between HyperNova and SuperNova.

In this study, HyperNova and SuperNova were fused and expressed with various molecules, but all of them were highlighted more brightly with HyperNova than with SuperNova. In this experiment, for example, HyperNova showed a great improvement in highlighting the p53, c-jun and Aurora A fusion molecules, which are known to have a fast turnover (approximately 30 min for p53[21], approximately 3.3 h for c-jun[22] and approximately 4 h for Aurora A[22]). This suggests that HyperNova should be ideal for the manipulation of proteins with fast turnover. For example, HyperNova could be used to analyze the phenotype after transient inactivation and then analyze whether the phenotype recovers by over time. This could be done as part of a CALI to confirm its specificity. Recently developed techniques for comprehensive analysis of protein turnover have revealed the presence of many high turnover proteins in cells[22–24]. In particular, high turnover proteins were found especially among proteins related to the cell cycle and transcription. Indeed, as we showed, HyperNova enabled CALI against c-jun, p53, and Aurora A, whereas SuperNova failed to inactivate all of them (Fig. 2e, f, h). Furthermore, using HyperNova, we have successfully developed a CALI method for MAPK, a molecule with a different function and distribution from the previous three molecules. Based on these results, we conclude that HyperNova, with its fast maturation properties, can be used to perform CALI on many molecules, regardless of their characters. In the future, we expect the application of HyperNova to allow more molecules to be manipulated, and we hope that the CALI method will become a common experimental technique. For example, the CALI method using HyperNova is expected to be particularly useful for elucidating the physiological functions of molecules with spatiotemporal characteristics, such as ERK2, in which activation takes the form of a wave[25].

As mentioned earlier, the CALI method using antibodies can manipulate endogenous molecules, but it is necessary to obtain antibodies capable of CALI. Therefore, it is difficult to perform this technique in a high-throughput manner. In the case of genetically encoded photosensitizing fluorescent proteins, exogenous expression of fusion molecules has been commonly used, but with recent advances in genome editing, there are reports of photosensitizing fluorescent proteins being knocked in with CRISPR–Cas9[26,27], which facilitates the manipulation of endogenous molecules. Therefore, we intend to combine CRISPR–Cas9 and HyperNova for the optical inactivation of endogenous molecules with CALI. Approximately 20,000 proteins seem to be expressed in vivo[28], many of which are thought to exhibit spatiotemporal behavior. We expect HyperNova to be useful to prove the causal relationship between individual molecular dynamics and their physiological functions in the broad field of life sciences.

## Methods
### Random mutagenesis screening
SuperNova was amplified by PCR with primers containing 5'-BamHI and 3'-KpnI sites. The restricted products were inserted in-frame into the same site in pQE80L to generate SuperNova/pQE80L as the template for the first mutagenesis. The SuperNova/pQE80L vector was subjected to error-prone PCR by the Diversify PCR Random Mutagenesis Kit (Clontech) with pQE80L forward and reverse primers. The mutation rates were 1.4-5.8/1000 bp. Mutated DNA fragments were shuffled as previously reported[29]. Briefly, DNA fragments were digested by DNaseI and reconstructed by PCR with pQE80L forward and reverse primers. The PCR product was then digested by BamHI/KpnI and inserted into the pQE80 BamHI/KpnI site.

The BL21 (DE3) was transformed with ligated solution by electroporation, and bacteria were grown on the nitro cellulose membrane (0.45 μm pore, GVS North America) attached to the LB plate containing 0.1 mg/mL ampicillin. On the next day, the membranes were moved to an IPTG-containing LB plate to induce protein expression. Two and a half hours after IPTG induction, when SuperNova did not yet show fluorescence, the colony emitting the strongest fluorescence was cultured overnight and used for plasmid purification. The resulting vector was used as template DNA in subsequent mutagenesis screening.

In the case of the second mutagenesis in ferritin fusion proteins, the synthetic ferritin gene and SuperNova1.5 gene from the final step of the first screening were inserted in-frame into BamHI and HindIII in pQE80L to generate the ferritin-SuperNova1.5/pQE80L vector, which was used as the template for the second mutagenesis. The DNA sequence was verified by dye terminator cycle sequencing using Big Dye (Applied Biosystems).

### Gene construction for E. coli expression
HyperNova and SuperNova-S10R genes were inserted into pQE80L using the BamHI and KpnI sites to generate HyperNova/pQE80L and Super-Nova-S10R/pQE80L, respectively. To construct the expression vectors for protein purification, HyperNova, SuperNova, SuperNova-S10R and EGFP genes were amplified by PCR with primers containing 5'-BamHI and 3'-EcoRI sites. The restricted products were inserted in-frame into the same site in pRSETb to generate HyperNova/pRSETb, SuperNova/pRSETb, Super-Nova-S10R/pRSETb and EGFP/pRSETb.

### Gene construction for mammalian expression in the cytosol
HyperNova/pRSETb, SuperNova/pRSETb and SuperNova-S10R/pRSETb were digested at BamHI and EcoRI sites, and the digested products were transferred to pcDNA3 using the same site to generate HyperNova/pcDNA3, SuperNova/pcDNA3 and SuperNova-S10R/pcDNA3, respectively.

### Gene construction for mammalian expression in mitochondria
Expression vectors in mitochondria, 2MLS-HyperNova/pcDNA3, 2MLS-SuperNova/pcDNA3, 2MLS-SuperNova-S10R/pcDNA3 and 2MLS-mSEGFP/pcDNA3 were constructed by fusing a tandem repeat of the mitochondria localization signal from cytochrome c oxidase subunit VIII[7] to the N-terminus of HyperNova, SuperNova, SuperNova-S10R and mSEGFP, respectively.

### Gene construction for mammalian expression of fusion molecule
The c-jun gene was amplified with primers containing 5'-BamHI and 3'-EcoRI sites. The HyperNova and SuperNova genes were amplified with primers containing 5'-EcoRI and 3'-XhoI sites. To add the linker sequence, the 5' primers also contained the three amino acid sequence GGS. The restricted products were inserted in-frame using the BamHI/XhoI sites of pcDNA3 to yield c-jun-HyperNova/pcDNA3 and c-jun-SuperNova/pcDNA3.

The HyperNova and SuperNova genes were amplified with primers containing 5'-BamHI and 3'-NotI sites. The restricted products were inserted using the BamHI/NotI sites of GFP-p53/N1 to yield HyperNova-p53/pcDNA3 and SuperNova-p53/pcDNA3.

The HyperNova and SuperNova genes were amplified with primers containing 5'-AgeI and 3'-BglII sites. The restricted products were inserted in-frame into the GFP-Aurora A/C1 vector to yield HyperNova-Aurora A/C1 and SuperNova-Aurora A/C1. The restricted products were inserted using the BamHI/KpnI sites of pcDNA3 by blunt-end ligation to yield HyperNova-Aurora A/pcDNA3 and SuperNova-Aurora A/pcDNA3.

The HyperNova and SuperNova genes were amplified by primers containing 5'-AgeI and 3'-NotI. The restricted products were inserted in-frame using the AgeI/NotI sites of the pActinin1-EGFP-N1 vector to yield Actinin-HyperNova/N1 and Actinin-SuperNova/N1.

The HyperNova gene was amplified by primers containing 5'-HindIII and 3'-BamHI. To add the linker sequence, the 5' primers also contained the

eight amino acid linker sequence GGSGGGSG. The restricted products were inserted in-frame using the HindIII/BamHI sites of SuperNova-fibrillarin/pcDNA3 to yield HyperNova-fibrillarin/pcDNA3.

The HyperNova and SuperNova genes were amplified by primers containing 5'-KpnI and 3'-EcoRI. To add the linker sequence, the 5' primers also contained the eight amino acid linker sequence GGSGGGSG. The restricted products were inserted in-frame using the KpnI/EcoRI sites of Aurora B-HaloTag7/pcDNA3 to yield Aurora B-HyperNova/pcDNA3 and Aurora B-SuperNova/pcDNA3.

The HyperNova gene was amplified by primers containing 5'-EcoRI and 3'-NotI. To add the linker sequence, the 5' primers also contained the 16 amino acid linker sequence as in pKeratin-SuperNova. The restricted products were inserted in-frame using the EcoRI/NotI sites of pKeratin-SuperNova to yield pKeratin-HyperNova.

Keratin-HyperNova/C1 was digested at BamHI and NotI sites and transferred to EB3-SuperNova/C1 using the same sites to generate EB3-HyperNova/C1.

The HyperNova gene was amplified by primers containing 5'-NheI and 3'-AccIII. The restricted product was inserted in-frame using the NheI/AccIII sites of SuperNova-RhoCA/C1 to yield HyperNova-RhoCA/C1.

The HyperNova gene was amplified by primers containing 5'-BamHI and 3'-XhoI. The restricted product was inserted in-frame using the BamHI/XhoI sites of Lyn-SuperNova/pcDNA3 to yield Lyn-HyperNova/pcDNA3.

The HyperNova gene was amplified by primers containing 5'-BamHI and 3'-NotI. To add the linker sequence, the 5' primer also contained the same 16 amino acid linker sequence as in pVimentin-SuperNova. The restricted products were inserted in-frame using the BamHI/NotI sites of pVimentin-SuperNova to yield pVimentin-HyperNova.

The HyperNova gene was amplified by primers containing 5'-NheI and 3'-XhoI. To add the linker sequence, the 3' primer also contained the 9 amino acid linker sequence GGSGGSGGS. The restricted products were inserted in-frame using the NheI/XhoI sites of pSuperNova-Tubulin to yield pHyperNova-Tubulin.

The cyclin A, SuperNova and HyperNova genes were amplified by primers. To add the linker sequence, the 3' primer of cyclin A and the 5' primer of SuperNova and HyperNova contained the 4 amino acid linker sequence GGSG. These amplified products were inserted in-frame at the EcoRV site of pcDNA3 to generate cyclinA-SuperNova/pcDNA3 and cyclinA-HyperNova/pcDNA3.

The cyclin D1, SuperNova and HyperNova genes were amplified by primers. To add the linker sequence, the 3' primer of cyclin D1 and the 5' primer of SuperNova and HyperNova contained the 4 amino acid linker sequence GGSG. These amplified products were inserted in-frame at the EcoRV site of pcDNA3 to generate cyclinD1-SuperNova/pcDNA3 and cyclinD1-HyperNova/pcDNA3.

The cyclin D2, SuperNova and HyperNova genes were amplified by primers. To add the linker sequence, the 3' primer of cyclin D2 and the 5' primer of SuperNova and HyperNova contained the 4 amino acid linker sequence GGSG. These amplified products were inserted in-frame at the EcoRV site of pcDNA3 to generate cyclinD2-SuperNova/pcDNA3 and cyclinD2-HyperNova/pcDNA3.

The plasmids for expressing c-jun (Addgene plasmid # 40348), p53 (Addgene plasmid # 12091), Aurora A (Addgene plasmid # 99877), Actinin1 (Addgene plasmid # 11908), cyclinA (Addgene plasmid # 8959), cyclinD1 (Addgene plasmid # 19927) and cyclinD2 (Addgene plasmid # 8950) were gifted from Dr. Alexander Dent, Dr. Tyler Jacks, Dr. Marc Tramier, Dr. Carol Otey, Dr. Bob Weinberg, Dr. Yue Xiong and Dr. Philip Hinds, respectively.

### Gene construction for mammalian expression in CALI for JNK1 and ERK2

The HyperNova gene was amplified by primers containing 5'-AgeI and 3'-XhoI. To add the linker sequence, the 3'-primer also contained the eight amino acid linker sequence GGSGGGSG. The restricted products were inserted in-frame using the AgeI/XhoI sites of EGFP-JNK1/C1 to yield

HyperNova-GGSGx2-JNK1/C1. The JNK1 gene was amplified by primers containing 5'-AgeI and 3'-BamHI, and the digested product was inserted using the AgeI/BamHI sites of GFP-JNK1/C1 to yield JNK1/pcDNA3. The amplified JNK-KTR and mSEGFP genes were inserted in-frame at the BamHI and EcoRI sites in pcDNA3 to generate JNK-KTR-mSEGFP/pcDNA3.

The ERK2 gene was amplified by primers containing 5'-KpnI and 3'-BamHI. The 3' primer also contained the eight amino acid linker sequence GGSGGGSG. The HyperNova gene was amplified by primers containing 5'-BamHI and 3'-XbaI. These two genes were inserted in-frame using the KpnI and XbaI sites of pcDNA3 to yield HyperNova-GGSGx2-ERK2/pcDNA3. The ERK2 gene was amplified by primers containing 5'-KpnI and 3'-XbaI, and the digested product was inserted using the KpnI/XbaI sites of pcDNA3 to generate ERK2/pcDNA3. The amplified ERK-KTR and mSEGFP genes were inserted in-frame using the BamHI and EcoRI sites in pcDNA3 to generate ERK-KTR-mSEGFP/pcDNA3. The plasmids of JNK1 (Addgene plasmid # 86830) and ERK2 (Addgene plasmid # 118296) were gifted from Dr. Rony Seger and Dr. Philip Stork, respectively. The plasmids encoding JNK-KTR (Addgene plasmid # 59151) and ERK-KTR (Addgene plasmid # 59150) were gifted from Dr. Markus Covert.

### Protein purification and size-exclusion chromatography

Purification of recombinant proteins with N-terminal polyhistidine tags was carried out as previously described[7] with some modifications. In brief, recombinant proteins were expressed in JM109 (DE3) at 23 °C for 4.5 days without IPTG induction by gentle shaking at 80 rpm. The polyhistidine-tagged proteins were purified by Ni-NTA column chromatography (Invitrogen) followed by buffer exchange to 50 mM HEPES-KOH (pH 7.4) with NAP-5 Column (Cytiva). To test the oligomeric state of HyperNova, size-exclusion chromatography was performed with a Superdex75 Increase 10/300GL column on ÄKTA go (Cytiva).

### ROS measurements in vitro

Singlet oxygen and superoxide measurements were performed as previously described[2]. Singlet oxygen was detected by anthracene-9,10-dipropionic acid (ADPA, Chemodex). Superoxide generation was measured by dihydroethidium (DHE, WAKO). Protein with ADPA or DHE was mixed in 50 mM HEPES-KOH (pH 7.4) to final concentrations of 50 µM and 10 µM, respectively. The samples were exposed to light provided by a 200 W mercury arc lamp (Hamamatsu Photonics) with a 580AF20 (Omega) filter (0.7 W/cm$^2$, 10 min). A heat-absorbing filter was installed between the mercury arc lamp and the excitation filter to prevent evaporation of the sample. After light exposure, the ADPA or DHE fluorescence was measured by a fluorescence spectrophotometer (FP-6200, Jasco).

### ROS measurements in living cells

To measure total ROS in mammalian cells, HeLa cells (RIKEN BRC, 0007) were transfected with pcDNA3 encoding SuperNova, SuperNova-S10R or HyperNova. Twenty-four h after transfection, the cells were incubated with 5 µM CellROX Green (Thermo Fisher) in DMEM without phenol red for 30 min at 37 °C. Then, the medium was removed, and the cells were washed 3 times with DMEM. The cells were irradiated by light (8 W/cm$^2$, 60 s) through a 580AF20 excitation filter (Omega) under an inverted microscope (IX83, Olympus) equipped with a UAPON 40 × 1.35 numerical aperture (NA) oil objective and a CMOS camera ORCA-SPARK (Hamamatsu). Before and after irradiation, the CellROX Green signal was detected and analyzed.

Superoxide measurement in the mitochondria of mammalian cells was performed as previously described[11]. HeLa cells were transfected with pcDNA3 encoding 2MLS-SuperNova, 2MLS-SuperNova-S10R or 2MLS-HyperNova. Twenty-four h after transfection, the cells were incubated with 2 µM MitoSOX (Thermo Fisher) in DMEM without phenol red for 10 min at 37 °C. Then, the medium was removed, and the cells were washed 3 times with DMEM. The cells were irradiated by light (4 W/cm$^2$, 30 s) through a 580AF20 excitation filter (Omega) under the same equipment to measure

total ROS in mammalian cells. Before and after irradiation, the MitoSOX signal was detected and analyzed.

## Bleaching experiment in living cells
HeLa cells were transfected with pcDNA3 encoding SuperNova or HyperNova. Twenty-four h after transfection, the medium was changed to DMEM without phenol red and exposed intense light through a 580AF20 excitation filter (Omega, without ND filter) under an inverted microscope with the same equipment as imaging experiment. The fluorescence image was detected for 5 min with 2 s intervals and examined the reduction of red fluorescence.

## Measurement of the phototoxic effect in E. coli and mammalian cells
To measure the phototoxic effect in *E. coli*, JM109 (DE3) was transformed with the pRSETb vector containing EGFP, SuperNova, SuperNova-S10R or HyperNova. The colony was cultured in 1 mL of LB medium for 15–16 h at 37 °C, and then 10 µL bacterial suspension was irradiated by light (1.3 W/cm$^2$, 2 min) through a 580AF20 excitation filter (Omega) using a 200 W mercury arc lamp as a light source. After plating the irradiated bacterial suspension, the numbers of surviving *E. coli* colonies with and without light irradiation were compared. To determine phototoxicity in mammalian cells, HeLa cells were transfected with pcDNA3 encoding 2MLS-EGFP, 2MLS-SuperNova, 2MLS-SuperNova-S10R or 2MLS-HyperNova. Twenty-four h after transfection, the cells were irradiated by light (6.7 W/cm$^2$, 90 s) through a 580AF20 excitation filter (Omega) and examined DIC time-lapse imaging under an inverted microscope (IX83, Olympus) equipped with a UPLSAPO 60 × 1.35 numerical aperture (NA) oil objective and a CMOS camera ORCA-SPARK (Hamamatsu). The number of surviving HeLa cells was counted at various time points after irradiation.

## Immunohistochemistry of HyperNova fusion protein
HeLa cells cultured on 35 mm glass bottom dishes were transfected with pcDNA3 encoding SuperNova or HyperNova fusion protein using a Fugene 6 Transfection Kit (Promega). The amount of transfected plasmid was 1.0 µg for p53, Aurora A, Aurora B and Actinin fusion vectors; 0.5 µg for Cyclin A, Cyclin D1 and Cyclin D2 fusion vectors; and 0.3 µg for c-jun fusion vector. After transfection for 52–53 h, the cells were fixed with 4% paraformaldehyde. Anti-KillerRed antibody (Evrogen, 2 µg/mL) and Alexa488-conjugated anti-rabbit IgG (ThermoFisher, 2 µg/mL) were used to detect fusion proteins. To calculate the ratio of fluorescence per protein expression, the fluorescence of photosensitizing protein and Alexa488 were detected by an IX83 fluorescence microscope.

## CALI experiments in transcriptional factors
HeLa cells cultured on 6 cm dishes were transfected with pcDNA3 encoding c-jun or p53 fused with SuperNova or HyperNova and each reporter plasmid using a Fugene 6 Transfection Kit. The amount of transfected plasmid in c-jun experiments was 0.6 µg for c-jun fusion, 0.2 µg for pJC6-GL3 (Addgene plasmid # 11979) and 2.2 µg for pcDNA3. The amount of transfected plasmid in p53 experiments was 1.5 µg for p53 fusion, 0.3 µg for PG13-Luc (Addgene plasmid # 16442) and 1.2 µg for pcDNA3. Twenty four h after transfection, medium was replaced by 0.2% FBS-DMEM, and the cells were incubated for 24 h before CALI to reduce the basal activities. Before light exposure, CALI samples were collected by centrifugation at 800 rpm, 5 min at room temperature. The samples were illuminated with light using the equipment described in the section on in vitro ROS measurement (1.27 W/cm$^2$, 2 min). After light irradiation, the cells were cultured in 10% FBS/DMEM for 6 h in c-jun and for 1.5 h in p53 to induce transcriptional activities. To evaluate the CALI effect, reporter assay was examined using One Glo luciferase assay kit (Promega).

## Live imaging and CALI of cell division
HeLa cells expressing EGFP-Aurora A, SuperNova-Aurora A or HyperNova-Aurora A were arrested at the G1/S transition using the double thymidine block method as previously described[2]. In brief, ten h after the second block, light irradiation was applied (7.9 W/cm$^2$, 1 min) to an IX83 inverted microscope (Olympus) equipped with a UPLSAPO 60×1.35 numerical aperture (NA) oil objective. To detect cell division, DIC imaging was started immediately after light irradiation.

## CALI experiment in JNK1
HeLa cells grown in 35 mm glass-bottom dishes were transfected with 0.75 µg pcDNA3 encoding HyperNova-GGSGx2-JNK1 and 0.25 µg JNK-KTR probe vector. Forty-eight to fifty-six h after transfection, the cells were incubated in serum-free DMEM for 4 hr and irradiated by light (7.9 W/cm$^2$, 60 s) through a 580AF20 excitation filter (Omega) under an inverted microscope (IX83, Olympus) equipped with a UPLSAPO 60×1.35 NA oil objective. After light irradiation, JNK activation was induced by the bath application of 5 µM anisomycin at the final concentration. JNK activity was analyzed as a ratio of GFP fluorescence in the nucleus and cytoplasm using the JNK-KTR probe, as previously described[16].

## CALI experiment in ERK2
COS7 cells[30] grown in 35 mm glass-bottom dishes were transfected with 0.75 µg pcDNA3 encoding HyperNova-GGSGx2-ERK2 and 0.25 µg ERK-KTR probe vector. Forty-eight to fifty-six h after transfection, the cells were incubated in serum-free DMEM for 4 hr and irradiated by light (7.9 W/cm$^2$, 60 s) to induce CALI. After light irradiation, ERK activation was induced by bath application of 50 ng/mL EGF at the final concentration. ERK activity was analyzed as the ratio of GFP fluorescence in the nucleus and cytoplasm using the ERK-KTR probe, as previously described[16]. The microscopic systems for CALI and imaging were the same as in the JNK1 experiments described above.

## Spectroscopy
To calculate the molar extinction coefficient of HyperNova, the absorbance of 10 µM protein solution was measured on a DU730 spectrometer (Beckman Coulter) at 579 nm. The molar extinction coefficient at 579 nm, $\varepsilon_{579}$ (M$^{-1}$ cm$^{-1}$), was calculated by the following equation: $\varepsilon_{579} = A_{579}/c$, where $A_{579}$ and c are the absorbance at 579 nm and the concentration of the sample (M), respectively. Relative fluorescence quantum yields of HyperNova (indicated as "sample", below) were calculated using the fluorescence quantum yields of SuperNova as a standard (indicated as "standard", below) as reported previously[31]. In brief, to determine the fluorescence spectra between 580 nm and 700 nm, a 0.5 µM protein sample was measured by a fluorescence spectrophotometer (571 nm excitation, FP-6200, Jasco). The relative fluorescence quantum yield was calculated by the following equation: $\phi_x = \phi_{st}$ x $(F_x/F_{st})$ x $(A_{st}/A_x)$ x $(n_x/n_{st})^2$. where $\phi_x$ and $\phi_{st}$ are the quantum yields of the sample and standard, respectively. $F_x$ and $F_{st}$ are the integrated fluorescence intensities of the sample and standard, respectively. $A_x$ and $A_{st}$ are the absorbance at 571 nm of the sample and standard, respectively. $n_x$ and $n_{st}$ are the refractive indices of the sample and standard, respectively. Note that the refractive indices of SuperNova and HyperNova were calculated to be the same.

## Imaging of fusion molecules in HeLa cells
HeLa cells cultured on 35 mm glass-bottom dishes were transfected with pcDNA encoding the HyperNova fusion protein using a Fugene 6 Transfection Kit (Promega). The amount of transfected plasmid was 1.0 µg. Forty-eight to fifty-six h after transfection, the expression of fusion proteins was observed under an inverted microscope (Ti-2, Nikon) equipped with a CFI Plan Apochromat λ 60 × 1.40 numerical aperture (NA) oil objective with a 1.5x zoom lens, a confocal unit (Dragonfly 200, Andor Technology), and an EMCCD camera (iXon Ultra, Andor Technology). We used a 561 nm laser for excitation and an emission filter, ET600/50 nm (Chroma).

## Statistics and reproducibility
All statistical analyses were performed with R programming software (ver. 4.2.2, R Development Core team). When the compared data sets included

nonnormal data sets, a nonparametric test, the Wilcoxon rank sum test, was performed (exactRankTests, ver. 0.8.35); otherwise, an unpaired t test was used after equality of variances was checked. All statistical tests were two-sided. In the analysis of singlet oxygen generation using ADPA (Supplementary Fig. 3c), only the data of HyperNova did not have normality, but the sample size was small. Thus, we applied unpaired t tests to the data sets. When multiple pairs of data sets were compared, *p* values were corrected using Bonferroni's method. For all box plots, the centerline indicates the median, the box indicates the range of the upper and lower quartiles, and the whiskers indicate the 1.5x interquartile range. If the observed data points were outside the range, they were plotted.

## Reporting summary

Further information on research design is available in the Nature Portfolio Reporting Summary linked to this article.

## Data availability

The nucleotide sequence of the HyperNova gene has been deposited in GenBank (LC771173). The data sets generated and/or analyzed during the current study are available from the corresponding author upon reasonable request. We deposited plasmids encoding HyperNova on Addgene (#221456: HyperNova / pRSETb and #221457: 2MLS-HyperNova / pcDNA3). All source data underlying the graphs presented in the main and Supplementary Figs. are uploaded as Supplementary Data.

## Code availability

We did not use custom algorithms or software in this study.

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

## Acknowledgements

We are grateful to Tomoki Matsuda at Osaka University for technical assistance in immunohistochemistry. This work was supported, in part, by grants from the JSPS KAKENHI Grant Numbers 21H00423, 21K19311, 22H02719 and 23K17408 (to K.T.); Canon Foundation (to K.T.); Nakatani Foundation (to K.T.); Mitsubishi Foundation (to K.T); Takeda Science Foundation (to K.T.); Naito Foundation (to T.N.);

Uehara Memorial Foundation (to T.N.); JSPS KAKENHI Grant Numbers 18H03987 and 18H05410 (to T. N.).

## Author contributions

H.S. performed the gene construction, CALI experiments, and data analysis and wrote the manuscript. T.S. assisted with the CALI experiments. S.J. assisted in the biochemical analysis. R.O.-N. and T.N. assisted in the microscopic analysis. K.T. performed the mutagenesis screening, gene construction, CALI experiments and biochemical analysis, the conceptual development, and wrote the manuscript.

## Competing interests

K.T. is the inventor on patent application JP2023/103027 submitted by Mie University and Yokohama City University that covers the development and application of HyperNova. The remaining authors declare no competing interests.
