## [Peer Review File · Communications Biology]

Reviewers' comments:

Reviewer #1 (Remarks to the Author):

This manuscript introduces HyperNova, a new efficient genetically encoded photosensitizer to perform Chromophore-Assisted Light Inactivation (CALI) for the acute and localized loss of protein function in living cells and tissue. The promise of CALI is tantalizing but as author points out, its use has been limited to a small number of applications (though they have been important and published in good journals). Genetically encoded photosensitizers have indeed caused a resurgence in CALI's application and improving these would enhance its general use and value to biology.

There has been an evolution from Killer Red (that requires dimerization) to SuperNova that functions for CALI as a Monomer to HyperNova. Here they use error prone PCR and DNA shuffling to generate a library of mutations in the SuperNova DNA and express this library in E. coli monitoring colonies for early fluorescence. They showed one mutant with 13 mutations now called HyperNova had increased rate of maturation and generated more singlet oxygen (though not superoxide) compared to SuperNova. It was tested for CALI efficiency alone using cell death as an assay. Their data show that it is about 2-fold better in ROS generation. Moreover, they can do CALI well for gene fusion proteins. This latter point is an important one increasing the proteins that CALI can be applied to. and with fusions to cjun, P53 and Aurora A using transcriptional reporters to assess loss of function. New CALI experiments using Jnk and Erk2, MAP kinases that act in stress response, and proliferation and migration respectively.

Overall, this is a well conducted set of experiments that support the authors claim of the usefulness of HyperNova over its predecessor, SuperNova. It should be of value extending the number of proteins that can be assessed for their cellular and in vivo function using CALI.

While perhaps beyond the scope of these experiments, the impact and value of HyperNova would have been better demonstrated to identify a novel finding of Erk2 function or perhaps a novel spatial or temporal requirement that could not otherwise be achieved except with CALI.

Minor points

Maturation is misspelled in the title

How do we know that maturation is the issue for SuperNova? They don't present data on this and no reference is cited that establishes this. It is also unclear what the mechanistic basis of fluorophore maturation and they should mention this. Speculate in the discussion on why the specific mutations promote maturation.

Should cell death induced by HyperNova be a concern for CALI experiments?

Are the fusions made with HyperNova or SuperNova 5' or 3'? This may make a difference in fusions if HyperNova folds rapidly and SuperNova slowly.

High turnover being an advantage is an interesting point. While they point this out as important to replace the endogenous protein with HyperNova-fused protein, fast turnover may also be helpful in recovery after light irradiation to address the dynamics of that proteins function. It also reduces concern of nonspecific damage or crosslinking simply gumming up the works. Seeing a time course of recovery after CALI for one of their examples would be useful.

RE P 8 "HyperNova is useful regardless of the fusion molecule in living cells, "

I think they have shown that it works better for a few molecules but given biological complexity, it still may not work for some specific protein so I would include some caution here. It would be interesting

to know if Light had effects on cells expressing HyperNova fusions to actin and tubulin as this would be expected.

Reviewer #2 (Remarks to the Author):

The paper reports a novel red genetically-encodable photosensitizer mutated from SuperNova with improved maturation efficiency at 37°C and in turn more efficient CALI inactivation in cells. While the advantages of the new variant (HyperNova) are convincingly shown and it is a useful addition to the CALI toolbox, there are several issues that need clarification:

-In several experiments, a comparison with EGFP is presented. However, this does not seem to make sense as EGFP and SuperNova/HyperNova are in completely different spectral regions, and a 580/20 bandpass filter is used to irradiate the samples, where EGFP does not absorb. Even if a suitable wavelength was used for EGFP, the results cannot be quantitatively compared as photons have different energies.

-What is the photostability of HyperNova compared to SuperNova? If more ROS is generated by the former, presumably photobleaching may also be higher.

-Supplementary Figure 3c shows that HyperNova degrades ADPA more efficiently than SuperNova, which is interpreted as higher singlet oxygen production efficiency by the former. However, it has been shown that ADPA is not a selective method for singlet oxygen determination in fluorescent proteins (J. Am. Chem. Soc. 2013, 135, 26, 9564). Indeed, while the original SuperNova paper reported an equivalent figure to Supplementary Figure 3c (but comparing SuperNova with KillerRed), it had been shown by direct detection of singlet oxygen that KillerRed does not produce singlet oxygen (Chem. Commun., 2011, 47, 4887, see also PLoS Biol 9(4): e1001041.; Free Rad. Biol. Med, 128, 2018, 157). Therefore, the ADPA degradation of both SuperNova and HyperNova is unlikely related to singlet oxygen production. A more selective method should be used (or the discussion should be limited to total ROS or superoxide production).

- Related to the above, the discussion about the change in photophysical properties in pages 9-10 should include the fact that while ROS production seems to increase in HyperNova, its fluorescence quantum yield remains the same. Is this expected?

Minor comments:

-There is a typo in the title

- Page 9, line 34, do the authors mean "photophysical" instead of "optical"?

- The authors may want to rephrase the sentence "We initially thought that we could wait a long time for the maturation of SuperNova,..." since it sounds too colloquial. Also, the first sentence of the introduction is too generic.

Response to the specific comments of reviewers:

Reviewer #1:

1) While perhaps beyond the scope of these experiments, the impact and value of HyperNova would have been better demonstrated to identify a novel finding of Erk2 function or perhaps a novel spatial or temporal requirement that could not otherwise be achieved except with CALI.

We thank the reviewer for the important suggestion to highlight the advantages of the CALI method. Since this paper focused on the technical advances of CALI, that experiment is beyond the scope as the reviewer mentioned. Therefore, we have described it as one of the possible applications for ERK2 CALI in the discussion section (Page 10-L35 > Page 11-L2).

Minor comments

1) Maturation is misspelled in the title.

We corrected the misspell in the title.

2) How do we know that maturation is the issue for SuperNova? They don't present data on this and no reference is cited that establishes this.

We thank the reviewer for the important remarks about the introduction of this paper. It has been reported in a previous study (Gorbachev DA et al. 2020) that there is potential for improvement in the folding of SuperNova. We have also noticed in some cases that SuperNova takes a long time to color, especially at 37°C, compared to other fluorescent proteins, as shown for example in Fig. 2 and Fig. S5. To clarify this point, we have modified the introduction section (Page 3-L20 > L23).

3) It is also unclear what the mechanistic basis of fluorophore maturation and they should

mention this. Speculate in the discussion on why the specific mutations promote maturation.

We thank the reviewer's comment to discuss the rule of these mutations to promote maturation. According to the reviewer's suggestion, we confirmed the distribution of each mutation in the structure of SuperNova and discussed the function of these mutations in the discussions section (Page 9-L35 > Page 10-L6).

4) Should cell death induced by HyperNova be a concern for CALI experiments?

As you can see from the CALI experiments with JNK and ERK, we were able to inactivate the target molecules with HyperNova without affecting molecules other than the target molecules (Fig. 3c-d and Fig. 4c-d). Therefore, we believe that CALI with HyperNova does not cause nonspecific damage that induces cell death and that it is not a concern in CALI experiments. However, if the irradiation light is too strong, we think it is necessary to be cautious because phototoxicity is expected to occur regardless of the expression of photosensitizing proteins.

5) Are the fusions made with HyperNova or SuperNova 5' or 3'? This may make a difference in fusions if HyperNova folds rapidly and SuperNova slowly.

In each experiment, whether HyperNova was fused to the 5' or 3' side was noted in the methods section. For example, in Fig. 2 and Supplementary Fig. 5, AuroraA and p53 were fused with the photosensitizing proteins at the 5' of the target molecules. On the other hand, c-jun, Actinin, AuroraB, CyclinA, CyclinD1, and CyclinD2 were fused with the photosensitizing proteins at the 3' of the target molecules. We have tried both 5' and 3' fusions and confirmed that HyperNova has a higher maturation ability than SuperNova under all conditions. Therefore, it does not seem to matter which end of HyperNova is fused to.

6) *High turnover being an advantage is an interesting point. While they point this out as important to replace the endogenous protein with HyperNova-fused protein, fast turnover may also be helpful in recovery after light irradiation to address the dynamics of that proteins function. It also reduces concern of nonspecific damage or crosslinking simply gumming up the works. Seeing a time course of recovery after CALI for one of their examples would be useful.*

We thank for the important suggestion. We agree with the reviewer's comment regarding CALI of fast turnover proteins. Indeed, we can transiently disrupt their function by HyperNova, take some time to recover it and see if the phenotype returns to normal. We have described this as the application of CALI in HyperNova in the discussion section (Page 10-L22 > L24).

7) *RE P 8 “HyperNova is useful regardless of the fusion molecule in living cells, “ I think they have shown that it works better for a few molecules but given biological complexity, it still may not work for some specific protein so I would include some caution here. It would be interesting to know if Light had effects on cells expressing HyperNova fusions to actin and tubulin as this would be expected.*

We agree with the reviewer's comments. Certainly, there are many proteins expressed *in vivo*, and we can expect that some molecules cannot be successfully manipulated by CALI with HyperNova. While CALI for actin and tubulin is interesting, we would like to modify our sentence to be more cautious here (Page 8-L6 > L7).

Reviewer #2:

1) *In several experiments, a comparison with EGFP is presented. However, this does not seem to make sense as EGFP and SuperNova/HyperNova are in completely different spectral regions, and a 580/20 bandpass filter is used to irradiate the samples, where EGFP does not absorb. Even if a suitable wavelength was used for EGFP, the results cannot be quantitatively compared as photons have different energies.*

We used EGFP to check the effects of irradiating the light itself in the experimental system,

such as phototoxicity on cells or secondary effects on ROS indicators. Therefore, we preferred to use molecules that do not absorb light as a negative control. To clarify this point, we have added sentences in the result section (Page 6-L14 > L16).

2) What is the photostability of HyperNova compared to SuperNova? If more ROS is generated by the former, presumably photobleaching may also be higher.

We sincerely appreciate this important suggestion. Indeed, in the bleaching experiment, only fluorescent molecules, that is, folded molecules, can be analyzed, so it is possible to determine whether the total ROS production ability itself has certainly increased regardless of the difference of the folding ability between HyperNova and SuperNova. We performed the bleaching experiment in living cells expressing HyperNova or SuperNova at 37°C. We confirmed that the photobleaching of HyperNova tended to be occurred rather lower than that of SuperNova. This suggests that the improved total ROS production at 37°C in HyperNova results not from an improved ability of ROS production, but from improved maturation at 37°C.

To accurately describe the characteristic of HyperNova, we would like to append this result and modify the sentences related to this point (Page 5-L20 > Page 6-L2, Supplementary Fig. 3c). We also revise the discussion (Page 10-L8 > Page 10-L15) and add the methods (Page 16-L16 > Page 17-L22) and legend (Supplementary Information Page 5-L8 > L10) for this experiment.

3) Supplementary Figure 3c shows that HyperNova degrades ADPA more efficiently than SuperNova, which is interpreted as higher singlet oxygen production efficiency by the former. However, it has been shown that ADPA is not a selective method for singlet oxygen determination in fluorescent proteins (J. Am. Chem. Soc. 2013, 135, 26, 9564). Indeed, while the original SuperNova paper reported an equivalent figure to Supplementary Figure 3c (but comparing SuperNova with KillerRed), it had been shown by direct detection of singlet oxygen that KillerRed does not produce singlet oxygen (Chem. Commun., 2011, 47, 4887, see also PLoS Biol 9(4): e1001041.; Free Rad. Biol. Med, 128, 2018, 157). Therefore, the ADPA degradation of both SuperNova and HyperNova is unlikely related to singlet oxygen production. A more selective method should be used (or

the discussion should be limited to total ROS or superoxide production).

We appreciate the reviewer's comment that raised an important point. We agree that we need to be cautious about the specificity of the ADPA. According to the reviewer's suggestion, we would like to remove the result of ADPA (Supplementary Fig. 3c). In relation to this, we also removed the discussion about the photophysical properties of HyperNova which is relevant to the ADPA experiment (Page 10).

4) Related to the above, the discussion about the change in photophysical properties in pages 9-10 should include the fact that while ROS production seems to increase in HyperNova, its fluorescence quantum yield remains the same. Is this expected?

Fluorescence quantum yields were measured on purified proteins expressed in *E. coli* at 18°C with sufficient maturation. We expected that if only the folding efficiency at 37°C was simply improved, it is possible that the fluorescence quantum yields of HyperNova could be the equivalent to that of SuperNova. Based on our results shown in Supplementary Fig. 3a-d including the bleaching experiment, we suggest that HyperNova has a higher folding efficiency, resulting in a higher percentage of molecules folding in the cell, which leads to an increase in apparent ROS production especially at 37°C. To clarify our view, we modified the result section (Page 5-L11 > L14) and the discussion section (Page 10-L8 > L15).

Minor comments:

1) There is a typo in the title

We corrected the misspell in the title.

2) Page 9, line 34, do the authors mean "photophysical" instead of "optical"?

As reviewer pointed out, we corrected "optical" to "photophysical".

3) *The authors may want to rephrase the sentence “We initially thought that we could wait a long time for the maturation of SuperNova, ...” since it sounds too colloquial.*

As reviewer pointed out, we have revised this statement to appropriate sentences for a thesis (Page 3-L20 > L23).

4) *Also, the first sentence of the introduction is too generic.*

As reviewer suggested, we remove the first sentence of the introduction. According to this modification, we also change the second sentence (Page 3-L2 > L4).

REVIEWERS' COMMENTS:

Reviewer #1 (Remarks to the Author):

The authors respond adequately to my issues and I approve publishing this manuscript. Here are a couple of corrections for clarity of the added text.

Page 3, Line 21 : Replace "it should never be matured" with "it doesn't mature"

Page 5, Line 11-12 Replace "may not be" with "is not"

Reviewer #2 (Remarks to the Author):

While the authors have addressed the issues raised in my previous report, the manuscript lacks clarity regarding the photophysical characterization:

- The authors compare experiments in cells with experiments performed with purified (and completely mature) protein and conclude that the apparent improved photophysical properties in cells are mostly due to improved maturation. This is presented in a rather confusing way, it would be clearer if the photophysical parameters in solution were established first, and describe later the experiments in cells.
- The authors should explain better that EGFP is a suitable control because it does not absorb at the wavelength used.
- Page 5, line 8, the authors probably mean "spectroscopic" instead of "stereoscopic".
- In general, the manuscript may benefit from English editing, especially the sentences added during the latest revision.

Response to the specific comments of reviewers:

Reviewer #1:

1) Page 3, Line 21 : Replace “it should never be matured” with “it doesn’t mature”

As reviewer pointed out, we replaced “it should never be matured” with “it doesn’t mature.” (Page 3-L21)

2) Page 5, Line 11-12 Replace “may not be” with “is not”

As reviewer pointed out, we replaced “may not be” with “is not.” (Page 5-L10)

Reviewer #2:

1) *The authors compare experiments in cells with experiments performed with purified (and completely mature) protein and conclude that the apparent improved photophysical properties in cells are mostly due to improved maturation. This is presented in a rather confusing way, it would be clearer if the photophysical parameters in solution were established first, and describe later the experiments in cells.*

In accordance with your suggestion, we changed the order of the explanations (Page 5-L14>Page6-L2).

2) *The authors should explain better that EGFP is a suitable control because does not absorb at the wavelength used.*

We added the sentence to explain that EGFP does not absorb at the wavelength used. (Page 6-L15-17)

3) Page 5, line 8, the authors probably mean “spectroscopic” instead of “stereoscopic”.

As the reviewer pointed out, we corrected “stereoscopic” to “spectroscopic.” (Page 5-L7)

4) *In general, the manuscript may benefit from English editing, especially the sentences*

added during the latest revision.

As reviewer suggested, we have carefully re-edited the English of the manuscript.